# TomoGRAF: An X-ray physics-driven generative radiance field framework for extremely sparse view CT reconstruction

**Di Xu[1], Yang Yang[2], Hengjie Liu[3], Qihui Lyu[1], Martina Descovich[1], Dan Ruan[3], Ke Sheng[1]***

1 Radiation Oncology, University of California, San Francisco, California, United States of America, 2 Radiology, University of California, San Francisco, California, United States of America, 3 Radiation Oncology, University of California, Los Angeles, California, United States of America

* ke.sheng@ucsf.edu

## Abstract

### Objectives

Computed tomography (CT) provides high spatial-resolution visualization of 3D structures for various applications. Traditional analytical/iterative CT reconstruction algorithms require hundreds of angular samplings, a condition may not be met practically for physical and mechanical limitations. Sparse view CT reconstruction has been proposed using constrained optimization and machine learning methods with varying success, less so for ultra-sparse view reconstruction. Neural radiance field (NeRF) is a powerful tool for reconstructing and rendering 3D natural scenes from sparse views, but its direct application to 3D medical image reconstruction has been minimally successful due to the differences in photon transportation and available prior information between optic and X-ray.

### Methods

We develop TomoGRAF to reconstruct high-quality 3D CT volumes using ultra-sparse projections. TomoGRAF has two main novelties pertinent to X-ray physics and CT imaging. First, TomoGRAF's volume rendering module accumulates x-ray material attenuation passing through an object with CT geometry rather than visible light material color and opacity from surface interaction in NeRF. Second, TomoGRAF penalizes the difference between the simulated and ground truth volume during training besides the 2D views, thus significantly improving the prior fidelity.

### Results

TomoGRAF is trained on LIDC-IDRI dataset (1011 scans) and evaluated on an unseen in-house dataset (100 scans) of distinct imaging characteristics from training

**Data availability statement:** Data cannot be shared publicly because of institutional restriction. Data are available from the UCSF Institutional Data Access / Ethics Committee for researchers who meet the criteria for access to confidential data. Interested researchers should follow the instructions outlined on https://icd.ucsf.edu/materialdata-transfer-agreements. Please contact Industrycontracts@ucsf.edu | (415) 350-5408 for additional assistance.

**Funding:** The research is supported by NIH R01CA259008, R44CA183390 and R01EB031577.

**Competing interests:** NO authors have competing interests.

and demonstrates a vast leap in performance compared with state-of-the-art deep learning and NeRF methods.

## Conclusion

TomoGRAF provides the first generalizable solution for image-guided radiotherapy and interventional radiology applications, where only one/a few X-ray views are available, but 3D volumetric information is desired.

## 1. Introduction

Computed tomography (CT) acquires x-ray projections around the subject to generate 3D cross-sectional images. Compared to 2D radiographs, where the depth information along the ray direction is lost and structures superimposed, CT enables the 3D representation of rich internal information for quantitative structure characterization. Analytically, Tuy's data sufficiency condition covering a sufficient sampling trajectory is required for mathematical rigid reconstruction [1]. Violating Tuy's condition leads to geometry and intensity distortion in the reconstructed images (referred to as limited-angle artifacts in the following). Besides the sampling trajectory, a minimal sampling density is required to avoid streak artifacts that can severely corrupt the image with sparse, e.g., < 100 views.

On the other hand, data-sufficient conditions may not always be met due to practical limitations, including imaging dose considerations, limited gantry freedom, and the need for continuous image guidance for radiotherapy and interventional radiology [2,3]. For instance, the total ionizing radiation exposure in mammograms is kept low to protect the sensitive tissue [4]. However, 2D mammograms without depth differentiation can be inadequate with dense breast tissues. Digital breast tomosynthesis (DBT), a limited-angle tomographic breast imaging technique, was introduced to overcome the problem of tissue superposition in 2D mammography while maintaining a low dose level. In DBT, limited projection views are acquired while the X-ray source traverses along a predefined trajectory, typically an arc spanning an angular range of 60° or less. The acquired limited angle samplings are then reconstructed as the volumetric representation with improved depth differentiation but still inferior quality to CT [5,6]. In different applications, the acquisition angles are not restricted, but the density of projection is reduced to lower the imaging dose [7,8] or to capture the dynamic information in retrospectively sorted 4DCT [9,10], resulting in sparse views in each sorting bin.

Image reconstruction from sparse-view and limited-angle samplings is an ill-posed inverse problem. Due to insufficient projection angles, the conventional filtered back-projection (FBP) [11] algorithm, algebraic reconstruction technique (ART) [12], and simultaneous algebraic reconstruction technique (SART) [13] suffer from limited-angle artifacts that worsen with sparser projections. Over the past few decades, substantial effort has been made to advance the development of sparse-view CT from two general avenues. One line of research lies in developing

regularized iterative methods based on the compressed sensing (CS) [14] theory. For instance, Sidky et al. proposed the adaptive steepest descent projection onto convex sets (ASD-POCS) method by minimizing the total variation (TV) of the expected CT volume from sparsely sampled views [15]. Following that, the adaptive-weighted TV (awTV) model was introduced by Liu et al. for improved edge preservation with local information constraints [16], while an improved TV-based algorithm named TV-stokes-projection onto Convex sets (TVS-POCS) was proposed immediately after to eliminate the patchy artifacts and preserving more subtle structures [17]. Apart from the TV-based methods, the prior image-constrained CS (PICCS) [18], patch-based nonlocal means (NLM) [19], tight wavelet frames [20], and feature dictionary learning [21] algorithms were introduced to further improve the reconstruction performance in representing patch-wise structure features. More recently, deep learning (DL) techniques were explored for improved CT reconstruction quality in the image or sinogram domain. The image domain methods learn the mapping from the noisy sparse-view reconstructed CT to the corresponding high-quality CT using diverse network structures such as feed-forward network [22,23], U-Net [24], and ResNet [25]. The sinogram domain methods work on improving/mapping the FBP algorithm [26–29] or interpolating the missing information in the sparse-sampled sinograms [30–33] with DL techniques. These and other deep learning-based sparse view CT reconstruction studies are comprehensively reviewed in Podgorsak et al. and Sun et al. [34,35].

Yet, with the exception where the same patient's different CT was used as the prior [36], little progress was made in reconstructing high-quality tomographic imaging using less than ten projections, a practical problem in real-time radiotherapy or interventional procedures. For the former, an onboard X-ray imager orthogonal to the mega-voltage (MV) therapeutic X-ray provides the most common modality of image-guided radiotherapy (IGRT). However, a trade-off must be made between slow 3D cone beam CT (CBCT) and fast 2D X-rays [37,38]. A similar trade-off exists in interventional radiology [39]. When real-time interventional decisions need to be made in the time frame affording one to two 2D X-rays, yet 3D visualization of the anatomy is desired, a unique class of ultra-sparse view CT reconstruction problems combining extremely limited projection angles *and* sparsity is created.

With the advancement of deep learning and more powerful computation hardware, several recent studies proposed harnessing inversion priors through training data-driven networks for single/dual-view(s) image reconstruction. Specifically, Shen et al. designed a three-stage convolutional neural network (CNN) trained on patient-specific 4DCT to infer CT of a different respiratory phase using a single or two orthogonal view(s) [40]. Ying et al. built a generative adversarial network framework (X2CT-GAN) with a 2D to 3D CNN generator to predict tomographic volume from two orthogonal projections [41]. Though promising, their generalization and robustness to external datasets have not been demonstrated and may be fundamentally limited by two factors: First, they generate volumetric predictions purely from 2D manifold learning. As a result, these networks are incapable of comprehending the 3D world and the projection view formation process [42]. Second, a prerequisite for deep networks with complex enough parameters to implicitly represent 2D to 3D manifold mapping is large and diverse training data, a condition difficult to meet for medical imaging [43].

An effective perspective to mitigate those problems is to leverage intermediate voxel-based representation in combination with differentiable volume rendering for a 3D-aware model, which requires smaller data to generalize. The Neural Radiance Fields (NeRF) [44] model successfully implemented this principle for volumetric scene rendering. NeRF proposed synthesizing novel views of complex scenes by optimizing an underlying continuous volumetric scene function using a sparse set of input views. NeRF achieved this by representing a scene with a fully connected deep network with the input of 5D coordinates representing the spatial location, view direction, and the output of the volume density and view-dependent emitted radiance at the spatial location. The novel view was synthesized by querying 5D coordinates along the camera rays and using volume rendering techniques to project the object surface color and densities onto an image [44]. NeRF was designed to generate unseen views from the same object and typically required fixed camera positions as supervision. As an improvement, GRAF, a 3D-aware generative model 2D-supervised by unposed image patches, introduced a conditional radiance field generator trained within the Generative Adversarial Network (GAN) framework [45] that is capable of rendering views of novel objects from given sparse projection views [42].

The success of NeRF and GRAF motivated their applications to solve the 3D tomography problems. MedNeRF [46] was proposed by Corona-Figueroa et al. for novel view rendering from a few or single X-ray projections. MedNeRF inherits the general GRAF framework, remains 2D-supervised, and assumes visible-light photon transportation configuration in the generator with an addition of self-supervised loss to the discriminator. However, there are distinct differences between CT volume reconstruction and "natural object" 3D representation rendering in terms of the available choices of training supervision, imaging setup, and properties of the rays. Specifically, 3D training supervision (object mesh with information on surface color) is often hard to acquire for "natural objects." In contrast, existing CT scans are an ideal volumetric training ground truth (GT) for fitting a 3D tomographic representation learning model. Moreover, optical raytracing works by computing a path from an imaginary camera (eye) through each pixel in a virtual screen and calculating the surface color and opacity of the object visible through the virtual screen via simulating ray reflection, shading, or refraction on the object surface (Fig 1(b)). The solving target of optical ray tracing is the object surface color $(r, g, b)$ and density $\sigma$ in a 3D location $(x, y, z)$ Meanwhile, x-rays are transported from the focal spot and pass through an object to the detector plane, accounting for scattering and attenuation (Fig 1(c)). The goal of CT reconstruction is voxel-wise material density $\delta$ at a 3D location $(x, y, z)$. Because of these major differences between natural scenes and 3D medical images, the direct application of NeRF has not resulted in usable CT with ultra-sparse views.

To overcome the challenges in ultra-sparse view CT reconstruction while maintaining the superior NeRF 3D structure representation efficiency, we introduced an x-ray-aware tomographic volume generator, termed TomoGRAF, to simulate CT imaging setup and use CT and its projections for 3D- and 2D-supervised training. TomoGRAF is further enhanced with a GAN framework and computationally scaled with sub-volume and image patch GTs training. To the best of our knowledge, this is the first pipeline that informs the NeRF simulator with X-ray physics to achieve generalizable high-performing CT volume reconstruction with ultra-sparse projection representation.

## 2. Materials and methods

### 2.1. Problem formulation

As illustrated in Fig 1(d), we formulated the problem of 3D image reconstruction from 2D projection(s) into a GAN-based DL framework, including modules of generator G and discriminator equation, given a series of 2D projections $\boldsymbol{X}$ denoted as $\{X_1, X_2, \cdots, X_N\}$ where $X_i \in \mathbb{R}^2 = \mathbb{R}^{H \times W}$ for $i \in [1, \ N]$, $N$ is the number of available 2D projections, $H$ and $W$ is the projection height and width. Our modeling target was to form a $G$ that can predict the 3D volume $\hat{Y} \in \mathbb{R}^3 = \mathbb{R}^{D \times H \times W}$ ($D$ represents volume depth) where $G$ is supervised by $\boldsymbol{X}$ and 3D volume GT $Y \in \mathbb{R}^3 = \mathbb{R}^{D \times H \times W}$, and is penalized by $D$ to encourage optimal convergence.

### 2.2. TomoGRAF framework

TomoGRAF does not aim to optimize densely posed projections for rendering a single patient volume. Instead, it targets fitting a network for synthesizing new patient volume by learning on various unposed projections. Note that the generator and discriminator work on image patches and sub-volumes during training for better efficiency, whereas a complete patient volume is rendered at inference time. The detailed components in TomoGRAF are shown in Fig 1(f–h). In what follows, we elaborate on the model architecture.

**2.2.1. Generator.** Adapted from GRAF [42], Our generator consists of three main components: ray sampling, conditional generative radiance field, and projection rendering. Ray sampling modules render the x-ray paths in 3D that are associated with truth patch/sub-volume, and the conditional generative radiance field module predicts the material density from a 3D location along the rendered x-ray paths. Lastly, the projection rendering module obtains the 2D composition from the predicted volumetric material densities. Overall, the generator $G$ takes x-ray source setup matrix $\boldsymbol{K}$, view direction (pose) $\boldsymbol{\xi} = (\theta, \phi)$, 2D sampling pattern $v$, shape code $z_{sh} \in \mathbb{R}^{M_s}$ and appearance code $z_a \in \mathbb{R}^{M_a}$ as input, and predicts a size $M \times M$ CT projection patch $P' \in \mathbb{R}^2 = \mathbb{R}^{M \times M}$ ($M$ is a hyper-parameter defined by user; $M = 32$

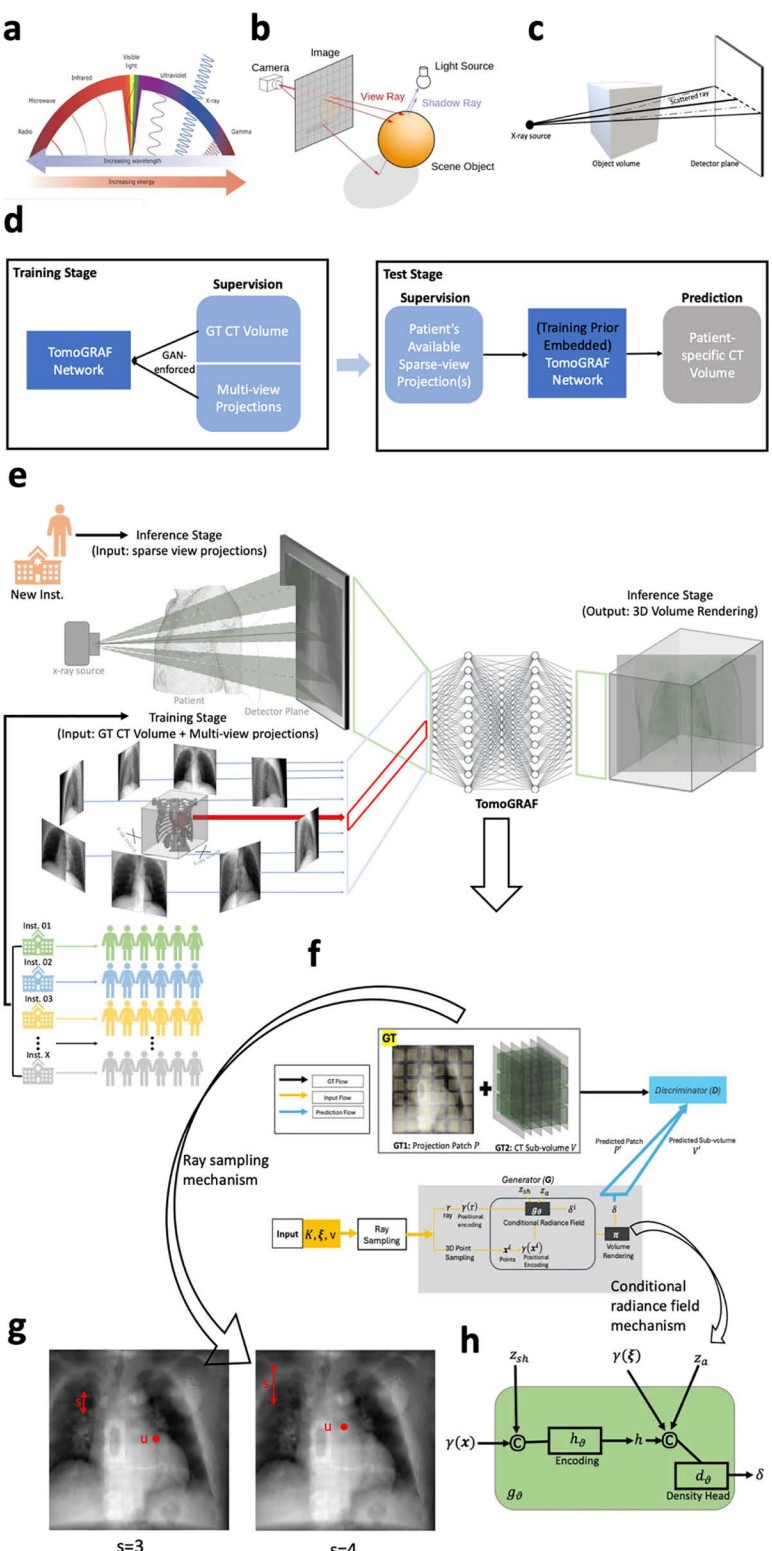

**Fig 1. Architecture of the proposed TomoGRAF framework.** a, The illustration of energy/wavelength difference between visible lights and X-rays. b, The visualization of object imaging with visible lights in 3D. c, The visualization of object imaging with X-ray in 3D. d, The diagram visualization of

the TomoGRAF pipline in the training and testing stages. **e,** The visualization of the training and inference stage of TomoGRAF. TomoGRAF references multi-view projections and the corresponding CT volume during training using data collected from multiple institutes (inst.) and will only require sparse-view projections referencing at the inference stage to render the predicted CT volume from a new institute. **f,** The pipeline of the TomoGRAF network. TomoGRAF works on projection patches and CT sub-volume in training. The input to TomoGRAF consists of source setup matrix $\boldsymbol{K}$, view direction (pose) $\xi$ and 2D sampling pattern $v$. **g,** The ray sampling mechanism in a patch of the projection input to TomoGRAF. $u$ represents the center of the sampled patch, and $s$ refers to the distance between two sampled patches. **h,** The design of conditional radiance field in TomoGRAF with a fully connected coordinate network ($g_\vartheta$) which consists of shape encoding $h_\vartheta$ and density head $d_\vartheta$.

is used in our experiments) and the associated CT sub-volume $V' \in \mathbb{R}^3$ corrsponding to $P'$ at $\xi$ (In orthogonal viewing, $V' \in \mathbb{R}^3 = \mathbb{R}^{M \times M \times D}$; In non-orthognal views, $V'$ has varied dimension and is essentially a collection of points where converging rays intersecting the 3D grid). $\boldsymbol{K}$ consists of $d = (d_1, d_2)$, $d_1$ is distance of source to patient (SAD), $d_2$ is distance of source to detector (SID). The detector resolution $S \in \mathbb{R}^2$ and $M$ is bounded by$(H, W)$.

**Ray Sampling**: The rendered rays $r \in \mathbb{R}^3$ are constrained by $\xi$, $\boldsymbol{K}$ and $P'$. The x-ray pose $\xi$ is sampled from a predefined pose distribution $p_\xi$ collected from projection angles in training data with the x-ray source facing towards the origin of the coordinate system all the time. As shown in Fig 1(g), $\boldsymbol{v} = (\boldsymbol{u}, s)$ determines the center $\boldsymbol{u} = (x', y\prime) \in \mathbb{R}^2$ and scales $s \in \mathbb{R}^+$ of $P'$that we target to predict, where $V'$ is formed with all the corresponding 3D coordinates that form $P'$. At the training stage, $\boldsymbol{u}$ and $s$ are uniformly drawn from $\boldsymbol{u} \sim U(\Omega)$ and $s \sim U[1, S]$ where $\Omega$ defined the 2D projection domain and $S = \min(H, W)/M$. Noteworthily, the coordinates in $P'$ and $V'$ are real numbers for the purpose of continuous radiance field evaluation. Following the stratified sampling approach in Mildenhall et al. [44], $Q$ number of points are sampled along each $r$. The number of rays $R = M \times M$ and $R = H \times W$ per training and inference, respectively.

**Conditional Generative Radiance Field**: Adapted from GRAF [42], the radiance field is represented by a deep fully connected coordinate network $g_\vartheta$ with the input of the positional encoding of a 7D vector $(x, y, z, \theta, \phi, z_{sh}, z_a)$ consisting of 3D location $\boldsymbol{x} = (x, y, z)$ and projection pose $\xi$. The output is the material density $\delta$ in the corresponding $\boldsymbol{x}$, where $\vartheta$ represents the network parameters and $z_{sh} \sim p_{sh}$ and $z_a \sim p_a$ with $p_{sh}$ and $p_a$ drawn from standard Gaussian distribution.

$$g_\vartheta : \mathbb{R}^{\mathcal{L}_{\boldsymbol{x}}} \times \mathbb{R}^{\mathcal{L}_\xi} \times \mathbb{R}^{M_{\boldsymbol{sh}}} \times \mathbb{R}^{M_{\boldsymbol{a}}} \to \mathbb{R}^3 \times \mathbb{R}^+ \tag{1}$$

$$(\gamma(\boldsymbol{x}),\ \gamma(\xi), z_{sh}, z_a) \to \delta \tag{2}$$

Where $\mathcal{L}_{\boldsymbol{x}}$ and $\mathcal{L}_\xi$ represent the latent codes of $\boldsymbol{x}$ and $\xi$, $M_{\boldsymbol{sh}}$ and $M_{\boldsymbol{a}}$ define the shape and appearance codes with $z_{sh} \in \mathbb{R}^{M_{sh}}$ and $z_a \in \mathbb{R}^{M_a}$, and $\gamma(\cdot)$ represents positional encoding.

The architecture of $g_\vartheta$ is visualized in Fig 1(h) with Equations (3–6). First, shape encoding $h_\vartheta$ is conducted with the input of $\gamma(\boldsymbol{x})$ and $z_{sh}$. Second, $h_\vartheta$ is concatenated with $\gamma(\xi)$ and $z_a$ and then sent to the density head $d_\vartheta$ to predict $\delta$.

$$h_\vartheta :\ \mathbb{R}^{\mathcal{L}_{\boldsymbol{x}}} \times \mathbb{R}^{M_{\boldsymbol{sh}}} \to \mathbb{R}^H \tag{3}$$

$$(\gamma(\boldsymbol{x}), z_{sh}) \to \boldsymbol{h} \tag{4}$$

$$d_\vartheta :\ R^H \times \mathbb{R}^{\mathcal{L}_\xi} \times \mathbb{R}^{M_{\boldsymbol{a}}} \to \mathbb{R}^1 \tag{5}$$

$$(\boldsymbol{h}, \gamma(\xi), z_a) \to\ \delta \tag{6}$$

where the encoding was implemented with a fully connected network with ReLU activation.

**X-ray Physics Informed Projection Rendering**: Lastly, given the material density $\{\delta_i^r\} = V\prime \in \mathbb{R}^3$ where $1 \leq i \leq Q$ of all points along the rays $\{r_j\}$ where $1 \leq j \leq M \times M$ in training, we used a CT projection algorithm, Siddon Ray Tracing algorithm [47], to synthetic 2D radiograph patch $P\prime$ given preset $K$ and $\xi$. In specific, Siddon algirhtm computes the path lenghs of the ray $r$ through each interescted voxel $x$ within the 3D grid to enable efficient ray path (line) intergral over the grid. The pseudo code for the applied Siddon algorithm is listed in S1 Appendix.

**2.2.2. Discriminator.** Following the discriminator architecture defined in MedNeRF [46], two self-supervised auto-encoded CNN discriminators, [48] $D_{1,\varnothing}$ and $D_{2,\varnothing}$ compare predicted sub-volume $V\prime$ to real sub-volume $V$ extracted from real volume $Y$ and predict projection patch $P\prime$ to real projection patch $P$ extracted from real projection $I$, respectively. $D_{1,\varnothing}$ is defined to convolve in 3D while $D_{2,\varnothing}$ is defined to convolve in 2D to align with the dimension of its discrimination targets. For extracting $V$ and $P$, we first extracted $P$ from a real projection $I$ given $v$ and $s$ randomly drawn from their corresponding distributions, and then located the coordinates of $V$ from $Y$ based on $\xi$ and $K$. $D_{1,\varnothing}$ and $D_{2,\varnothing}$ are backpropagated separately with their respective weights, while we defined them to share weights while discriminating different sub-volume/patch locations.

**2.2.3. Supervision.** A distinctive supervision strategy is designed for the training and testing phases to better adapt to the imaging setup and available information in CT reconstructions.

During training, the model is guided using hybrid (2D and 3D) supervision. Specifically, multiple 2D subpatches $P$ from different view angles and their paired subvolmes $V$ are used as GTs to converge the prior. This approach enables the model to efficiently and effectively establish a trained prior that represents the universal features shared across the training cohort. The use of 3D supervision is particularly advantageous in CT imaging, as it enables the model to learn not only the external surface but also the complex internal anatomy of the object. Unlike visible light-based reconstruction methods, which are typically limited to surface information from 2D views and are incapable of exploiting the full 3D volume, TomoGRAF leverages the availability of 3D ground truth data to guide the reconstruction of deeper anatomical structures.

The inference stage can be further divided into two sub-steps, including patient-specific fine tuning and CT volume prediction. For patient-specific finetuning, the trained prior is further optimized to adapt to the incoming patient's morphologies with supervision of the patient's 2D sparse-view projection. At this stage, we use the complete 2D projections instead of 2D subpatches for supervision. This approach prioritizes strict maintenance of global structures and anatomical consistency during fine-tuning, albeit with a higher demand for GPU memory. Following the fine-tuning process, where the trained prior is tailored to the specific morphologies of the patient using their sparse-view 2D projections, the algorithm reconstructs the complete 3D CT volume. This reconstructed volume incorporates patient-specific anatomical details, ensuring that the final output accurately represents the unique structural characteristics of the individual.

**2.2.4. Loss function. Training stage**: Our loss objective consists of discrimination towards patch as well as sub-volume predictions. First, the global structures in intermediate decoded patches of $D_{1,\varnothing}$ and $D_{2,\varnothing}$ were separately assessed by Learned Perceptual Image Patch Similarity (LPIPS) [49] (denoted in Equations (7–8)).

$$L_{r,V} = E_{f_v \sim D_1(v),\ v \sim V}[\frac{1}{whd} \left|\left| \varnothing_i(\mathcal{G}(\boldsymbol{f_v})) - \varnothing_i(\mathcal{T}(v)) \right|\right|_2]$$

(7)

$$L_{r,P} = E_{f_p \sim D_2(p),\ p \sim P}[\frac{1}{whd} \left|\left| \varnothing_i(\mathcal{G}(\boldsymbol{f_p})) - \varnothing_i(\mathcal{T}(p)) \right|\right|_2]$$

(8)

Where $\varnothing_i(\cdot)$ denotes the output from the $i$th layer of the pretrained VGG16 [50] network, $\boldsymbol{f_v}$ and $\boldsymbol{f_p}$ represent the feature maps from $D_{1,\varnothing}$ and $D_{2,\varnothing}$, $w, h$ and $d$ stands for the width, height and depth of the corresponding feature space, $\mathcal{G}$ is the pre-processing on $\boldsymbol{f}$, and $\mathcal{T}$ is the processing on truth sub-volumes/patches.

Second, hinge loss was selected to classify $P'$ from $P$ and $V\prime$ from $V$ with formulas listed in Equations (9–10).

$$L_{h,V} = E_{v\prime \sim V\prime} \left[ f(D_{1,\varnothing}(v')) \right] + E_{v \sim V} \left[ f(-D_{1,\varnothing}(v)) \right] \tag{9}$$

$$L_{h,P} = E_{p\prime \sim P\prime} \left[ f(D_{2,\varnothing}(p')) \right] + E_{p \sim P} \left[ f(-D_{2,\varnothing}(p)) \right] \tag{10}$$

Where $f(t) = \max(0,\ 1+t)$.

Lastly, data augmentation, including random flipping and rotation, was implemented to V' and P' prior to sending into $D_{1,\varnothing}$ and $D_{2,\varnothing}$, following the theory proposed by Data Augmentation Optimized for GAN (DAG) framework [51]. $D_{1,\varnothing}/D_{2,\varnothing}$ share weights while discriminating multiple augmented sub-volumes/patches. Therefore, we have the overall loss objective formulated as Equations (11–12).

$$L\left(\vartheta, \{\varnothing_{1,k}\}, \{\varnothing_{2,k}\}\right) = L\left(\vartheta, \varnothing_{1,0}, \varnothing_{2,0}\right) + \frac{\lambda_1}{n-1} \sum_{k=1}^{n} L\left(\vartheta, \varnothing_{1,k}, \varnothing_{2,k}\right) \tag{11}$$

$$L\left(\vartheta, \varnothing_{1,k}, \varnothing_{2,k}\right) = L_{r,V,\ k} + L_{h,V,k} + \lambda_2 \left( L_{r,P,\ k} + L_{h,P,k}\right) \tag{12}$$

Where $n = 4$, $\lambda_1 = 0.2$ and $k = 0$ corresponding to the identity transformation follows the definition in Trans et al [51]. $\lambda_2 = 0.5$ to give the model more attention on conformal $V\prime$ rendering.

**Inference Stage**: A fully trained $G_\vartheta$ was further fine-tuned with $z_{sh}$ and $z_a$ using the trained prior and incoming patients' sparse view projection(s) to render the final patient-specific volumetric prediction $\hat{Y}$. Since we conducted moderate optimization with limited iterations, $G_\vartheta$ was tuned with fully size $I$ instead of patches. Depending on the available views, a referenced projection was randomly drawn for each iteration until $G_\vartheta$ reached the convergence criteria. In our experiments, peak signal-to-noise ratio (PSNR)=25 was set as the stopping threshold. The inference loss objective is defined in Equation (13) with a combination of LPIPS, PSNR, and the negative log-likelihood loss (NLL).

$$L_{G_\vartheta} = \lambda_1 L_{r,I} + \lambda_2 L_{PSNR,I} + \lambda_3 L_{NLL,I} \tag{13}$$

Where $\lambda_1 = \lambda_3 = 0.3$ and $\lambda_2 = 0.1$ were set in our experiments.

## 2.3. Implementation details

During training, the RMSprop optimizer [52] with a batch size of 4 ($4 \times 1$), learning rate of 0.0005 for the generator, learning rate of 0.0001 for the discriminator, and 40000 iterations were performed. Per inference fine-tuning, RMSprop [52] optimizer with a batch size of 1, learning rate of 0.0005, stopping threshold of PNSR = 25 (mostly under 1000 iterations) was implemented towards the generator. All the experiments were carried out on a NVDIA RTX 4 × A6000 cluster.

## 2.4. Evaluation metrics

We evaluate the predicted CT volume $\hat{Y}$ and projection $\hat{I}$ corresponding to the reference view of our TomoGRAF generator using PSNR, structure similarity index measurement (SSIM) and rooted mean squared error (RMSE) as Equations (14–16).

$$PSNR = 20 \cdot \log_{10} \frac{MAX_I}{RMSE} \tag{14}$$

$$SSIM = \frac{(2\mu_{G_\vartheta}\mu_y + c_1)(2\sigma_{G_\vartheta y} + c_2)}{(\mu_{G_\vartheta}^2 + \mu_y^2 + c_1)(\sigma_{G_\vartheta}^2 + \sigma_y^2 + c_2)} \tag{15}$$

$$RMSE = \sqrt{\frac{\sum_{i=1}^{N} \|y(i) - \hat{y}(i)\|^2}{N}}$$

(16)

Where $MAX_I$ is the max possible pixel value in a tensor, RMSE stands for rooted mean squared error, $\mu_{G_\vartheta}$ and $\mu_y$ is the pixel mean of $G_\vartheta$ and $y$ and $\sigma_{G_\vartheta y}$ is the covariance between $G_\vartheta$ and $y$, $\sigma_{G_\vartheta}^2$ and $\sigma_y^2$ is the variance of $G_\vartheta$ and $y$. Lastly, $c_1 = (k_1 L)^2$ and $c_2 = (k_2 L)^2$, where $k_1 = 0.01$ and $k_2 = 0.03$ in the current work and $L$ is the dynamic range of the pixel values ($2^{\# \text{ bits per pixel}} - 1$).

## 2.5. Baseline algorithms

A CNN-based method X2CT-GAN [41], and a NeRF-based method MedNeRF [46] were included as our benchmarks with both the performance in projection inference and CT volume rendering compared. X2CT-GAN and MedNeRF are evaluated using the open-sourced codes and network weights released by authors, with the input of our in-house test set arranged following their data organization guidelines.

## 2.6. Data cohorts

1011 CT scans were selected from LIDC-IDRI [53] thoracic CT database for organizing the training set. Digital reconstructed radiographs (DRRs) were generated as projections for training supervision. Several scanner manufacturers and models were included (GE Medical Systems LightSpeed scanner models, Philips Brilliance scanner models, Siemens Definition, Siemens Emotion, Siemens Sensation, and Toshiba Aquilion). The tube peak potential energies for scan acquisition include 120 kV, 130 kV, 135 kV, and 140 kV. The tube current is in the range of 40–627 mA. Slice thickness includes 0.6 mm, 0.75 mm, 0.9 mm, 1.0 mm, 1.25 mm, 1.5 mm, 2.0 mm, 2.5 mm, 3.0 mm, 4.0 mm and 5.0 mm. SAD includes 595 mm, 541 mm, 570 mm and 535 mm with corresponding SID of 1085.6 mm, 949.1 mm, 1040 mm and 940 mm, respectively. The in-plane pixel size ranges from 0.461 to 0.977 mm [53]. 72 DRRs that cover a full 360° (generated each of 5° rotations) vertical rotations were generated for each scan. All the DRRs and CT volumes were black-border cropped out. DRRs were resized with a resolution of 128 times 128, and CT volumes were interpolated with a resolution of $\times H \times W = 128 \times 128 \times 128$ for model learning preparation. All the data were patient-wise normalized to [0, 1].

The test data was organized under IRB approval (IRB # 20–32527), entitled "Image-guided radiation therapy", which include 100 de-identified CT scans from lung cancer patients who underwent robotic radiation therapy. Informeed consent was not required for the retrospective imaging study. All patients were scanned by Siemens Sensation with tube peak potential energy of 120 kV, tube current of 120 mA, slice thickness of 1.5 mm, and in-plane pixel size of 0.977 mm. SAD and SID are of 570 mm and 1040 mm. The anterior-posterior (AP) and lateral views were generated for inference reference, with 1-view-based inference solely referencing the AP view projection. All the DRRs and CT volumes had the black border cropped out. Additionally, DRRs were resized with a resolution of $128 \times 128$, and CT volumes were interpolated with a volume size of $128 \times 128 \times 128$. All the data were patient-wise normalized to [0, 1] prior to being fed into models for inference.

## 2.7. Model performance as a function of the number of views and 3D supervision

The model baseline performance was established using a single AP view. 1, 2, 5, and 10 view reconstructions were also performed to determine the model performance. The view angles are specified as follows: for 1-view-based reconstruction, the AP view is used for referencing. For 2-view-based reconstruction, the AP and lateral views are used for inferencing. For 5-view-based reconstruction, a full 360° is covered with rotation every 72°, starting from the AP view. For 10-view-based reconstruction, a full 360° is covered with rotation of every 36°, starting from the AP view.

## 3. Results

The results of TomoGRAF, MedNeRF, and X2CT-GAN with 1 or 2 views for 2D projection and volume rendering are visually demonstrated in Figs 2 and 3, with accompanying statistics reported in Table 1 and Fig 4. TomoGRAF consistently outperforms MedNeRF and X2CT-GAN in both tasks with the most evident advantage in 1-view volume reconstruction.

For 2D projection rendering, TomoGRAF achieves marginally better results than MedNeRF. Both models maintain the overall critical body shapes of GT, and TomoGRAF visualizes more detailed morphology, such as heart, spine, and vascular structures. In comparison, the projection results of X2CT-GAN show visible distortion and significantly worse quantitative performance.

For 3D volume reconstruction, as shown in Fig 3, with 1-view, TomoGRAF depicts rich and correct anatomical details with visible tumors and a pacemaker consistent with GT. The results are further refined with the second orthogonal X-ray view, improving fine details' recovery. In comparison, MedNeRF and X2CT-GAN fail to render patient-relevant 3D volumes with 1 or 2 views. MedNeRF loses most anatomical details; X2CT-GAN deforms 3D anatomies that do not reflect patient-specific characteristics, such as lung tumors and the pacemaker.

As shown in Table 1, TomoGRAF is vastly superior in quantitative imaging metrics, achieving SSIM and PSNR of $0.79 \pm 0.03$ and $33.45 \pm 0.13$, respectively, vs. MedNeRF (SSIM at $0.37 \pm 0.05$ and PSNR at $7.68 \pm 0.10$) and X2CT-GAN

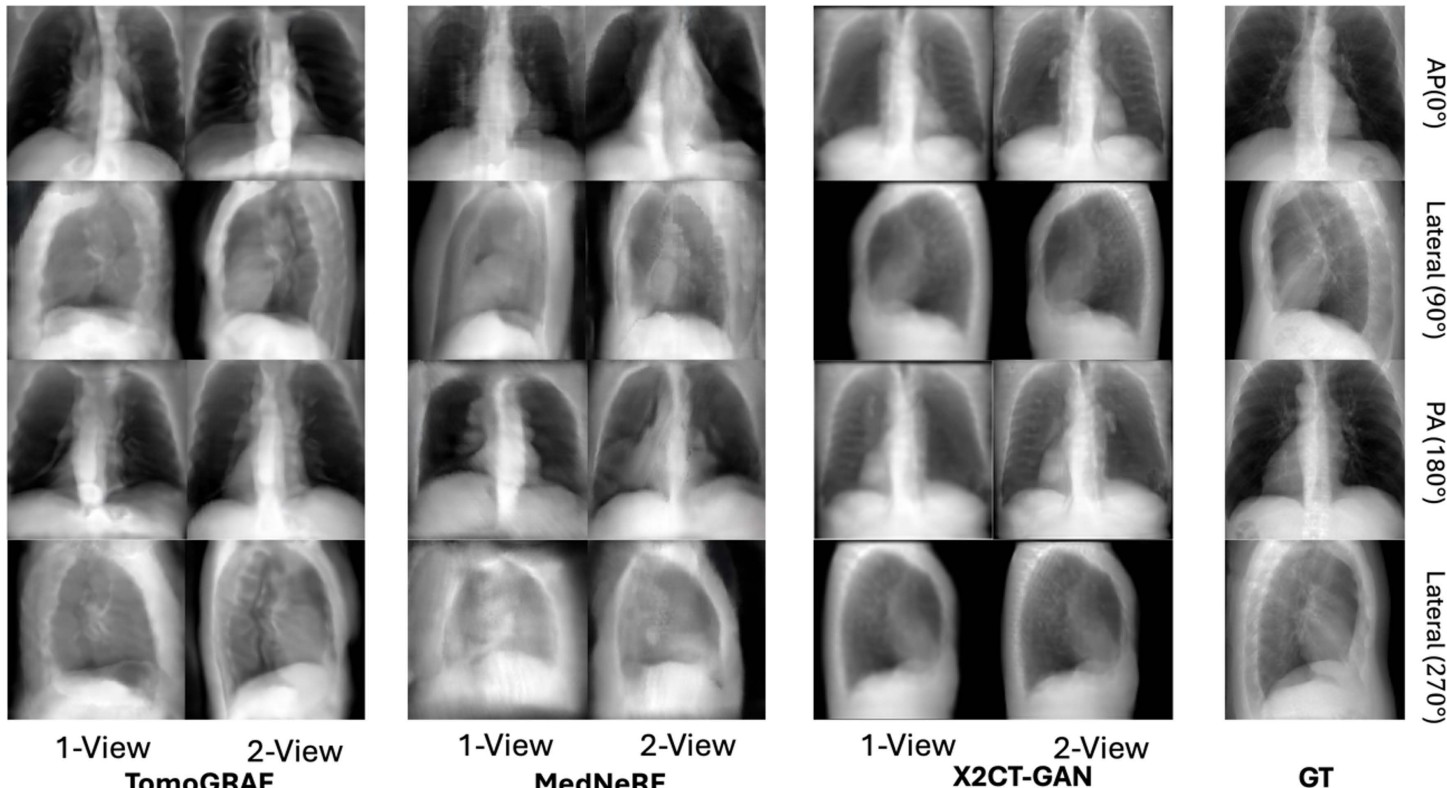

**Fig 2. Lung projection rendering in four 360°-clockwise-rotated (visualized every 90° of rotation) views from a patient in a test set.** Projections from X2CT-GAN are generated by applying the DRR synthesis algorithm on predicted CT volumes since the original X2CT-GAN was only designed to predict the CT volume. TomoGRAF and MedNeRF directly predicted the 2D projections. Results rendered by referencing 1-view and 2-views are both visualized. All the images are shown with a normalized window of [0, 1].

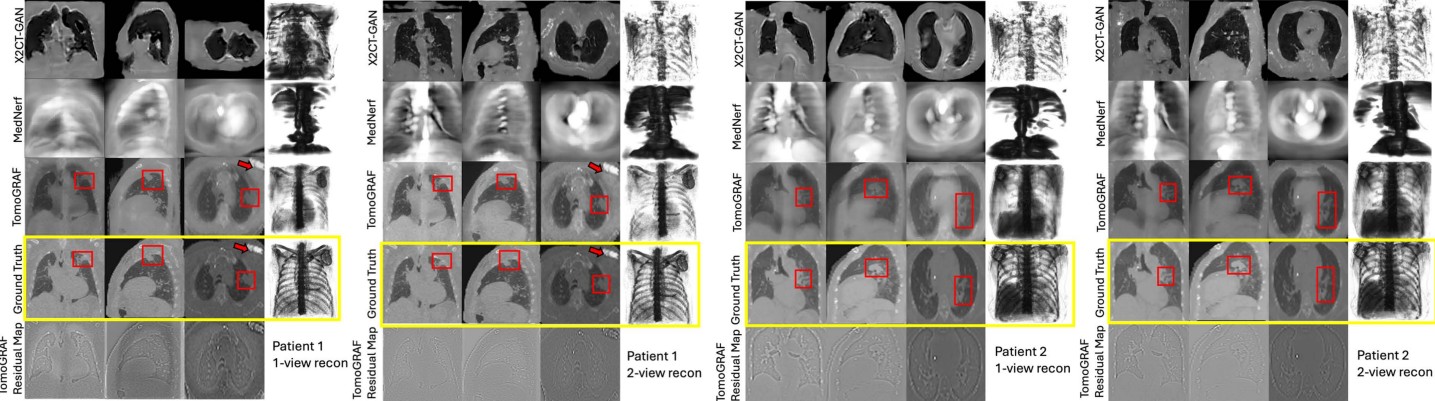

**Fig 3. CT Reconstruction results for two representative patients in the test set.** Only the residual maps of the TomoGRAF are shown, as the two comparison methods result in a large mismatch with GT. The images are visualized in a lung window with (a Hounsfield Unit width and level) of (1500 −600). The red boxes denote the lung tumors, while the arrows point to the pacemaker in patient 1.

**Table 1. Statistical results evaluated on the test set. The best results from each metric are underscored. ↑ indicates the higher the statistical value, the better, and vice versa for ↓. SSIM and PSNR are calculated with images normalized to [0,1] scales and RMSE are calculated based on Hounsfield Units (HUs). 0.31±0.12.**

|  | Modality | 1-View | | | | 2-Views | | | |
|---|---|---|---|---|---|---|---|---|---|
|  |  | SSIM↑ | PSNR(dB)↑ | RMSE(HU)↓ | Inference Time (s)↓ | SSIM↑ | PSNR(dB)↑ | RMSE↓ | Inference Time (s)↓ |
| CT Volume | X2CT-GAN | 0.31±0.12 | 14.39±0.19 | 386.69±27.43 | 0.27 | 0.48±0.06 | 17.35±0.21 | 347.76±25.46 | 1.31 |
|  | MedNeRF | 0.37±0.08 | 7.68±0.10 | 321.87±22.87 | 527.78±15.48 | 0.50±0.09 | 18.21±0.09 | 299.49±21.58 | 865.46±30.81 |
|  | TomoGRAF | 0.79±0.03 | 33.45±0.13 | 175.48±10.47 | 344.25±10.32 | 0.85±0.04 | 35.89±0.13 | 146.73±9.63 | 719.46±26.78 |
| Projection | X2CT-GAN | 0.34±0.11 | 11.88±0.19 | 51.96±7.98 | – | 0.51±0.09 | 18.23±0.26 | 47.64±7.32 | – |
|  | MedNeRF | 0.67±0.07 | 25.02±0.15 | 36.48±4.36 |  | 0.69±0.08 | 27.31±0.14 | 33.42±4.01 |  |
|  | TomoGRAF | 0.69±0.03 | 25.43±0.14 | 34.37±4.58 |  | 0.71±0.04 | 27.99±0.13 | 31.22±4.12 |  |

(SSIM at $0.31 \pm 0.012$ and PSNR at $14.39 \pm 0.19$). There is a similar reduction in RMSE. Additionally, we can observe from Fig 4 that the SSIM distribution of TomoGRAF is highly left skewed and leptokurtic in both 1 and 2-view-based volume rendering, with the majority clustering tightly towards the higher end, while that of MedNeRF and X2CT-GAN tends to be normal and moderately right-skewed (values lean towards the lower end).

Table 2 shows the 3D reconstruction performance with or without 3D supervision using 1, 2, 5, and 10 views as input. Using more views improved both volume and projection inference performance. 3D GT training markedly boosted the model performance in 3D volume rendering only. Fig 5 shows line profile comparisons for varying view inputs. 1 view TomoGRAF recovered major structures but missed fine details, which were better preserved with more X-ray views.

## 4. Discussion

This paper presents a GAN-embedded NeRF generator (TomoGRAF) for volumetric CT rendering from ultra-sparse X-ray views. TomoGRAF extends radiance fields into medical imaging reconstruction with a CT imaging-informed ray casting/ tracing simulator. Also, TomoGRAF leverages the availability of 3D volumetric information at the training stage to enable an effective generator trained with full volumetric supervision. The robustness of TomoGRAF is demonstrated on an external dataset independent of the training set. TomoGRAF vastly improves 1-view 3D reconstruction performance yet scales well with additional views to accommodate practical balances in image acquisitions and quality requirements.

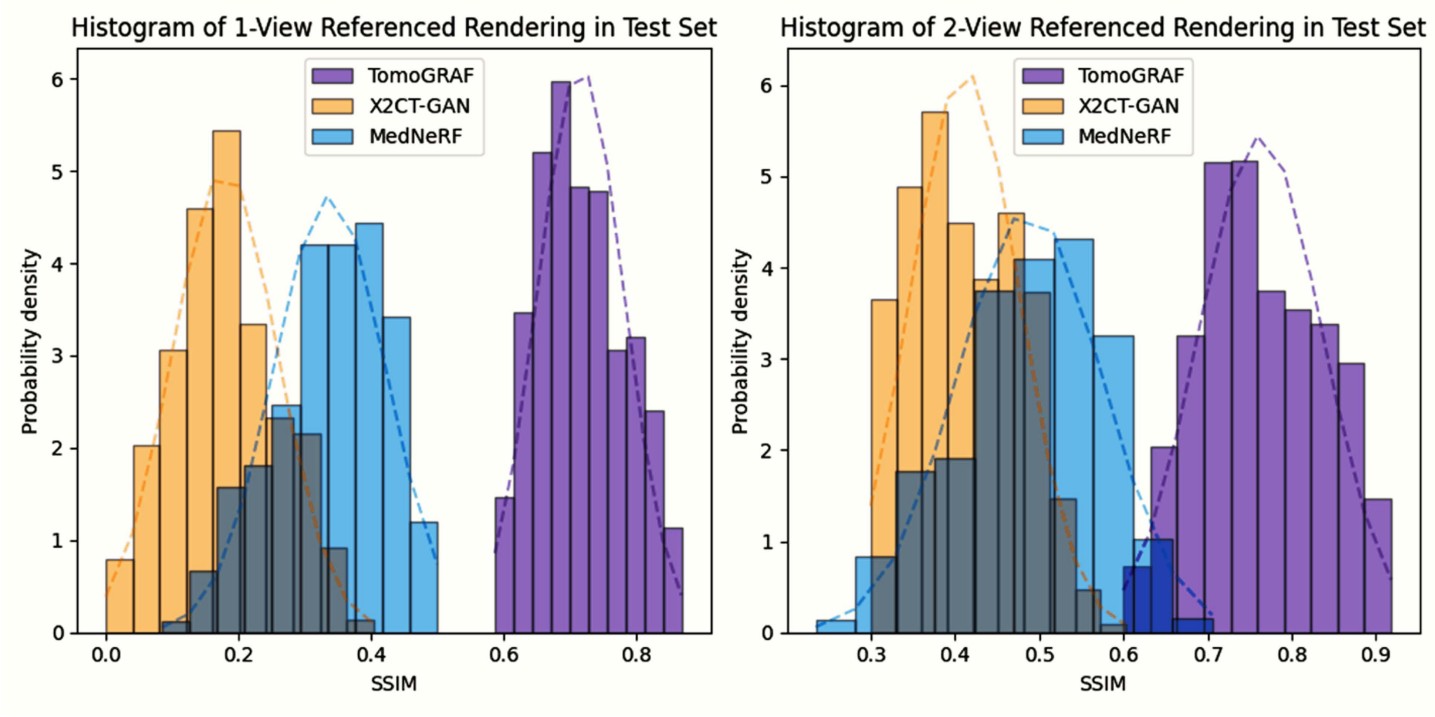

**Fig 4. Patient-wise evaluated SSIM distribution (visualized using histograms with smoothed trend curves) for a, 1-view-based volume rendering results of TomoGRAF, X2CT-GAN, and MedNeRF in the test set. b,** 2-view-based volume rendering results of TomoGRAF, X2CT-GAN, and MedNeRF in the test set.

**Table 2. Statistical results of TomoGRAF ablation study evaluated on test set. 1/2/5/10-Views represent the number of views used for reconstruction reference. Training with and without 3D CT supervision is also compared in the current table. ↑ indicates the higher the statistical value, the better, and vice versa for ↓. SSIM and PSNR are calculated with images normalized to [0,1] scales and RMSE are calculated based on Hounsfield units (HUs).**

| | Reference View | 3D Supervised Training | SSIM↑ | PSNR(dB)↑ | RMSE(HU) ↓ | Inference Time (s) ↓ |
|---|---|---|---|---|---|---|
| **CT Volume** | 1 | Y | 0.79±0.03 | 33.45±0.13 | 175.48±10.47 | 344.25±10.32 |
| | | N | 0.66±0.05 | 26.76±0.16 | 197.47±11.24 | |
| | 2 | Y | 0.85±0.04 | 35.89±0.13 | 146.73±9.63 | 719.46±26.78 |
| | | N | 0.69±0.06 | 29.87±0.19 | 168.35±10.21 | |
| | 5 | Y | 0.88±0.03 | 37.23±0.13 | 138.45±9.12 | 987.35±37.89 |
| | | N | 0.72±0.04 | 30.15±0.18 | 147.56±9.79 | |
| | 10 | Y | 0.93±0.01 | 39.98±0.11 | 127.68±8.78 | 1238.81±46.72 |
| | | N | 0.75±0.01 | 31.86±0.18 | 138.98±9.54 | |

Reconstruction of 3D CT volume from ultra-sparse angular sampling is an ill-posed inverse problem that is extremely underconditioned to solve. On the other hand, such 3D reconstruction is practically desirable and widely applicable when a full gantry rotation is prevented by mechanical limitations or the dynamic process of interest is significantly faster than CT acquisition speed [5,6,9]. Therefore, there has been a consistent effort to reconstruct 3D images with extremely sparse views that circumvent these mechanical and temporal restrictions. Although CS-powered iterative methods and some earlier DL methods were able to reconstruct 3D images with as few as 20 views [18], the resultant image quality

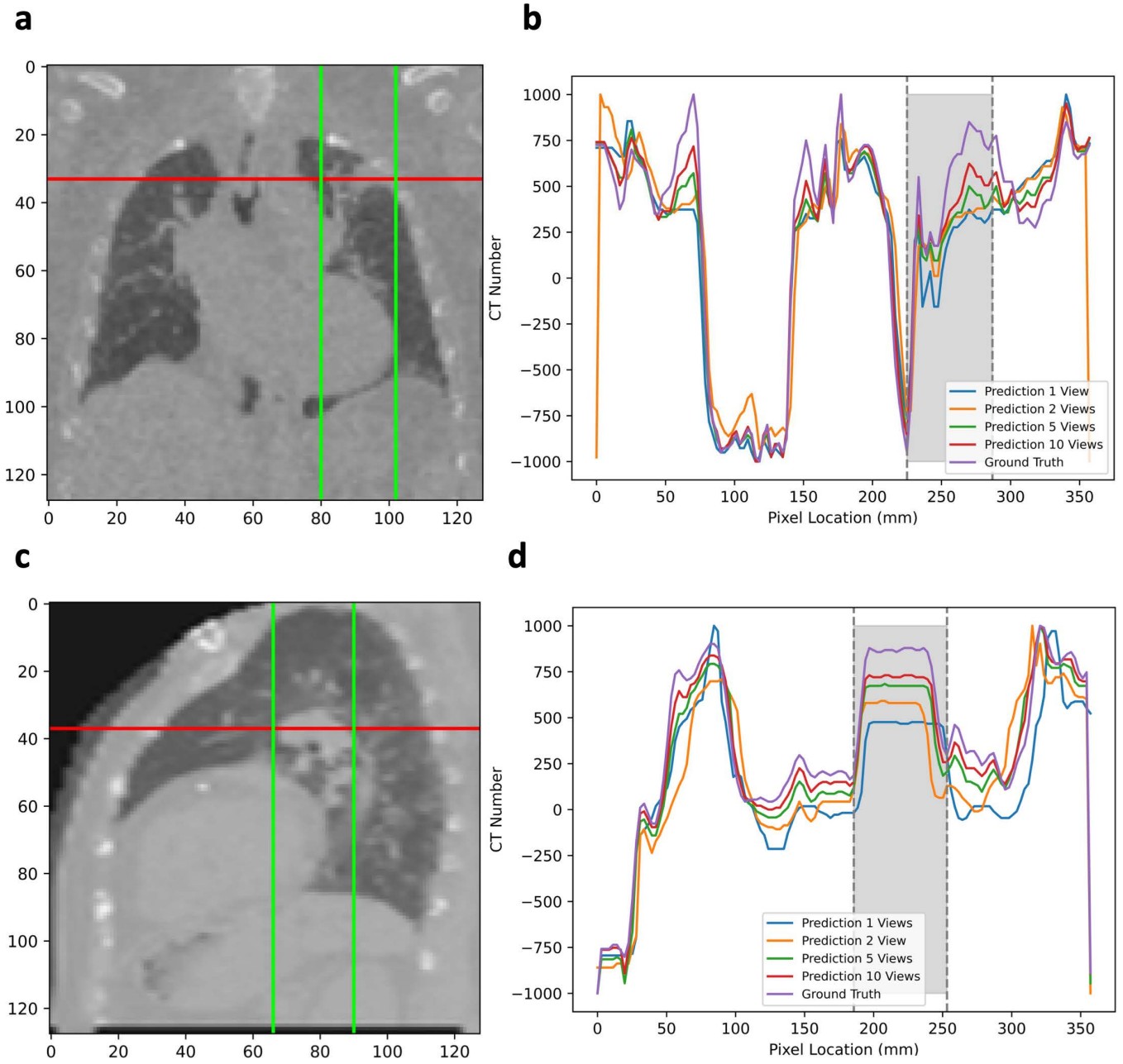

**Fig 5. Line profile comparison of the two patients shown in Fig 3.** The images visualized in a and **c** are GT slices downsampled to 128 × 128 to align with the prediction resolution. **a,** Indication of the location of the profile in patient 1 in coronal view across the lung tumor. **b,** Line profiles of 3D images rendered by TomoGRAF with 1, 2, 5, and 10-view inputs for patient 1. **c,** Indication of the location of the profile in patient 2 across the lung tumor in sagittal view. **d,** Line profiles of 3D images rendered by TomoGRAF with 1, 2, 5, and 10-view inputs for patient 2.

noticeably degraded. Still, they were unable to meet the challenges of many aforementioned practical scenarios where only one or two views were available for a given anatomical instance. Reconstruction with even more sparse views cannot be achieved without stronger priors and statistical learning. DL methods marched further in realizing ultra-sparse sampling (1 view) reconstruction using state-of-the-art (SOTA) networks, two of which are compared in this study.

TomoGRAF is distinctly superior to two SOTA methods for ultra-sparse view CT reconstruction in the following aspects. 1) In comparison to CNN-based networks with an extremely large number of parameters, such as X2CT-GAN [41], the radiance field-based generators train a lighter model with significantly fewer parameters (X2CT-GAN: ∼4.28G FLOPS vs. TomoGRAF ∼0.9G FLOPS, FLOPS: floating point operation per second) to represent the interaction process between photons and objects, which formalizes a better-defined goal for the network to reach and can effectively reduce the amount of training data required for achieving global robustness. Plus, CNN frameworks generally lack flexibility in view referencing. The number and angle of views at the inference stage must align with the input at the training stage. In addition to single-view referencing, TomoGRAF is capable of incorporating multiple X-ray views from diverse directions, as demonstrated in this study using uniformly distributed acquisition angles. It is worth noting that X2CT-GAN performance in the current study is markedly worse than the original report [41]. To determine the correctness of our implementation, we tested the X2CT-GAN code on the LIDC-IDRI data with the same data split and arrived at a similar performance. We believe the sharp decline in performance from internal LIDC-IDRI data testing to external in-house organized data testing is due to the differences in the training and testing data. CT images in LIDC-IDRI are cropped to keep only the thoracic organs, while our in-house test data are intact CT with the complete patient's chest wall, arms, and neck. Such variation or domain shift is common and expected in practice: the patients can vary in size and be set up with different immobilization devices or arm positions. In stark contrast to X2CT-GAN, TomoGRAF is robust to such variation. 2) Compared to MedNeRF, we adapt NeRF with a physically realistic volume rendering mechanism based on the x-ray transportation properties, where photons pass through the body, and the volumetric photon attenuation along the ray path within the body is the focus of reconstruction. In other words, TomoGRAF learns 3D X-ray image formation physics, whereas MedNeRF assumes visible light transportation physics and is intended for 2D manifold learning of object surfaces. The limitation is evident in both low quantitative imaging metrics and orthogonal cuts of MedNeRF reconstructed patients: there is better retention of outer patient contour than internal anatomical details, which are largely lost in MedNeRF images. Additionally, TomoGRAF employs paired 3D CT supervision at the training stage to maximize the prior knowledge exposed to the network, which contributed to the model robustness in volume rendering at the inference stage. As a result, TomoGRAF successfully leverages the efficient object representation capacity of NeRF while overcoming the intrinsic limitations due to its lack of X-ray transportation physics and 3D volume comprehension. To our knowledge, TomoGRAF is the first truly generalizable single-view 3D X-ray reconstruction pipeline robust to substantial domain shifts.

At the practical level, TomoGRAF provides a unique solution for applications where only one or a few X-ray views are available, but 3D volumetric information is desired. The applications include image-guided radiotherapy, interventional radiology, and angiography. For the former, 2D kV X-rays can be interlaced with MV therapeutic X-ray beams to provide a real-time view of the patient during treatment [54]. However, the 2D projection images do not describe the full 3D anatomy, which is critical for adapting radiotherapy to the real-time patient target and surrounding tissue geometry. Similarly, 4D CT digitally subtracted angiography (DSA) better describes dynamics of the contrast for enhanced diagnosis than single-phase CT DSA [55], but fast helical and flat panel-based 4D-DSA requires repeated scans of the subject, increasing the imaging dose and leading to compromised temporal resolution for intricate vascular structures [56]. TomoGRAF can be potentially used to infer real-time time-resolved 3D DSA with significantly reduced imaging dose. Our results show that TomoGRAF is flexible in incorporating more views for further improved inference performance. Dual views with fixed X-ray systems are widely used in radiotherapy for stereotactic localization [38,57,58], but the modality is limited to triangulating bony anatomies or implanted fiducials. TomoGRAF can utilize the same 2D stereotactic views to provide rich 3D anatomies for soft tissue-based registration and localization. Besides mechanical and imaging dose constraints, inexpensive portable 2D X-rays are more readily available for point-of-care and low-resource settings where a CT is impractical. The ability to reconstruct 3D volumes using a single 2D view would markedly

increase the imaging information available for clinical decisions. Our study also shows the feasibility of using more views in TomoGRAF for further improved performance and broader applications, including 4D CBCT and tomosynthesis with sparse or limited angle views.

At the theoretical level, TomoGRAF validates the extremely high data efficiency of neural field representation of 3D voxelized medical images. TomoGRAF, for the first time, materializes high data efficiency, achieving good quality (SSIM = 0.79–0.93) 3D reconstruction of CT images with 1–10 views, which is a major stride in comparison to existing research using NeRF or GRAF. The work thus has significant implications in 3D image acquisition, storage, and processing, which are currently voxel-based. Voxelized 3D representation does not provide intrinsic structural information regarding the relationships among voxels and thus can be expensive to acquire and reconstruct. Previous compressed sensing research explored some of the explicit structural correlations, such as piece-wise smoothness, for reduced data requirements. TomoGRAF indicates a new form of data representation that exploits implicit structural information with higher efficiency than conventional methods or neural networks without encoded physics.

Nevertheless, the current study leaves several areas for future improvement. First, TomoGRAF requires further fine-tuning at the inference stage, which increases the reconstruction time (1-view at $344.25 \pm 10.32$ s and 2-view at $719.46 \pm 26.78$ s). The time further increases with inference using more views. Significant acceleration is desired for online procedures such as motion adaptive radiotherapy [59]. Model compression techniques such as network pruning [60] and quantization [61] can decrease computational complexity while maintaining accuracy. Additionally, hardware acceleration via TensorRT [62] optimization or specialized processors (e.g., FPGAs [63], TPUs [64]) could also potentially improve the inference speed. Architecturally, incorporating efficient neural representations (e.g., lightweight MLPs [65] or hash-based encoding [66]) and adaptive sampling [67] methods could reduce computational overhead by prioritizing critical regions. Future work will explore these optimizations to improve TomoGRAF's feasibility for real-time clinical applications, which would be essential for interventional procedures. Second, TomoGRAF is developed and tested on CT-synthesized DRR, which differs from kV X-rays obtained using an actual detector in image characteristics due to simplification of the physical projection model, detector dynamic ranges, noise, pre and postprocessing [39]. The current model may need to be adapted based on actual X-ray projections. Third, TomoGRAF reconstruction results with 1-view are geometrically correct but lose fine details and CT number accuracy, which is partially mitigated with increasing views up to 10. Therefore, in its current form, TomoGRAF is suited for object detection and localization tasks, but its appropriateness for quantitative tasks such as radiation dose calculation needs to be further studied. Moreover, the recovery of detail should also improve with reconstruction resolution, which is currently limited in rendering a maximum of 128 times 128 times 128 resolution due to GPU memory constraints. This limitation, however, is expected to be overcome soon with rapidly-increasing GPU memory capacity. Additionally, TomoGRAF exhibits moderate interpretability, as its foundation in generative radiance fields aligns with ray-based CT physics, ensuring a degree of physical consistency. The use of 2D DRRs and paired 3D ground truth during training enhances structured learning, while the subpatch-based approach improves generalizability. However, the implicit representation of NeRF structure poses challenges in direct voxel interpretation. While GAN-based training further introduces a black-box component, inference remains L2-based, reducing the risk of unrealistic features. Sparse-view adaptation further complicates interpretability, as the model's implicit prior may influence reconstructions in ways distinct from traditional model-based iterative reconstruction methods. Future improvements could include feature sensitivity analyses and latent space visualization [68] to better delineate learned structures from data-driven priors. Lastly, while the current study primarily focuses on demonstrating the technical feasibility of ultra-sparse view reconstruction of the proposed TomoGRAF framework, we recognize that conventional quantitative image quality metrics may not fully capture the clinical utility of reconstructed images. As a future direction, incorporating clinical evaluation, such as qualitative scoring by radiologists or task-based diagnostic assessment, will further inform the real-world applicability and reliability of TomoGRAF in clinical practice.

## 5. Conclusion

TomoGRAF, a novel GAN-based NeRF generator, is presented in the current work. TomoGRAF is trained on a public dataset and evaluated on 100 in-house lung CTs. TomoGRAF reconstructed good quality 3D images with correct internal anatomies using 1–2 X-ray views, which state-of-the-art DL methods fail to accomplish. TomoGRAF performance further improves with more views. The superior TomoGRAF performance is attributed to novel x-ray physics encoding in the radiance field training and paired 3D CT supervision.

## Supporting information

**S1 Appendix. Siddon's Ray Tracing algorithm pseudo code applied in TomoGRAF projection rendering module.**
(DOCX)

## Author contributions

**Conceptualization:** Di Xu, Ke Sheng.

**Data curation:** Di Xu.

**Formal analysis:** Di Xu.

**Funding acquisition:** Ke Sheng.

**Investigation:** Di Xu.

**Methodology:** Di Xu.

**Project administration:** Ke Sheng.

**Resources:** Hengjie Liu, Ke Sheng.

**Supervision:** Ke Sheng.

**Validation:** Di Xu.

**Visualization:** Di Xu.

**Writing – original draft:** Di Xu, Hengjie Liu.

**Writing – review & editing:** Yang Yang, Hengjie Liu, Qihui Lyu, Martina Descovich, Dan Ruan, Ke Sheng.

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
