## [Decision Letter · Decision Letter 0]

13 Jun 2025

PONE-D-25-15335TomoGRAF: An X-Ray Physics-Driven Generative Radiance Field Framework for Extremely Sparse View CT ReconstructionPLOS ONE

Dear Dr. Sheng,

Thank you for submitting your manuscript to PLOS ONE. After careful evaluation, the reviewers raised some relevant concerns that need to be addressed first. Therefore, we invite you to submit a revised version of the manuscript that addresses the points raised during the review process.

We look forward to receiving your revised manuscript.

Kind regards,

Zhentian Wang, Ph.D.

Academic Editor

PLOS ONE

**Journal Requirements:**

1. When submitting your revision, we need you to address these additional requirements. Please ensure that your manuscript meets PLOS ONE's style requirements, including those for file naming. The PLOS ONE style templates can be found at https://journals.plos.org/plosone/s/file?id=wjVg/PLOSOne_formatting_sample_main_body.pdf and https://journals.plos.org/plosone/s/file?id=ba62/PLOSOne_formatting_sample_title_authors_affiliations.pdf 2. Thank you for stating in your Funding Statement: The research is supported by NIH R01CA259008, R44CA183390 and R01EB031577.  Please provide an amended statement that declares *all* the funding or sources of support (whether external or internal to your organization) received during this study, as detailed online in our guide for authors at http://journals.plos.org/plosone/s/submit-now.  Please also include the statement “There was no additional external funding received for this study.” in your updated Funding Statement. Please include your amended Funding Statement within your cover letter. We will change the online submission form on your behalf. 3. Thank you for stating the following in the Acknowledgments Section of your manuscript: The research is supported by NIH R01CA259008, R44CA183390 and R01EB031577. We note that you have provided funding information that is not currently declared in your Funding Statement. However, funding information should not appear in the Acknowledgments section or other areas of your manuscript. We will only publish funding information present in the Funding Statement section of the online submission form. Please remove any funding-related text from the manuscript and let us know how you would like to update your Funding Statement. Currently, your Funding Statement reads as follows: The research is supported by NIH R01CA259008, R44CA183390 and R01EB031577.  Please include your amended statements within your cover letter; we will change the online submission form on your behalf. 4. We note that you have indicated that there are restrictions to data sharing for this study. For studies involving human research participant data or other sensitive data, we encourage authors to share de-identified or anonymized data. However, when data cannot be publicly shared for ethical reasons, we allow authors to make their data sets available upon request. For information on unacceptable data access restrictions, please see http://journals.plos.org/plosone/s/data-availability#loc-unacceptable-data-access-restrictions.  Before we proceed with your manuscript, please address the following prompts: a) If there are ethical or legal restrictions on sharing a de-identified data set, please explain them in detail (e.g., data contain potentially identifying or sensitive patient information, data are owned by a third-party organization, etc.) and who has imposed them (e.g., a Research Ethics Committee or Institutional Review Board, etc.). Please also provide contact information for a data access committee, ethics committee, or other institutional body to which data requests may be sent. b) If there are no restrictions, please upload the minimal anonymized data set necessary to replicate your study findings to a stable, public repository and provide us with the relevant URLs, DOIs, or accession numbers. Please see http://www.bmj.com/content/340/bmj.c181.long for guidelines on how to de-identify and prepare clinical data for publication. For a list of recommended repositories, please see https://journals.plos.org/plosone/s/recommended-repositories. You also have the option of uploading the data as Supporting Information files, but we would recommend depositing data directly to a data repository if possible. Please update your Data Availability statement in the submission form accordingly. 5. Please include your full ethics statement in the ‘Methods’ section of your manuscript file. In your statement, please include the full name of the IRB or ethics committee who approved or waived your study, as well as whether or not you obtained informed written or verbal consent. If consent was waived for your study, please include this information in your statement as well.

Reviewers' comments:

Reviewer's Responses to Questions

**Comments to the Author**

1. Is the manuscript technically sound, and do the data support the conclusions?

Reviewer #1: Yes

Reviewer #2: Yes

2. Has the statistical analysis been performed appropriately and rigorously? 

Reviewer #1: Yes

Reviewer #2: N/A

3. Have the authors made all data underlying the findings in their manuscript fully available?

Reviewer #1: Yes

Reviewer #2: Yes

4. Is the manuscript presented in an intelligible fashion and written in standard English?

Reviewer #1: Yes

Reviewer #2: Yes

5. Review Comments to the Author

**Reviewer #1:**  This article describes a machine learning framework to allow for 3D medical image reconstruction from limited angle viewing data. While my background includes experience with NeRF and related techniques, I do not have a medical background. Nonetheless, I enjoyed reading this article, understood its significance, and am pleased to recommend it for publication. I have a few comments that, if properly addressed, will substantially improve the paper:

1) My understanding of the method, as described in Sec. II, relied heavily on reading the background literature related to GRAF, MedNeRF and X2CT-GAN. I highly recommend that the authors consider creating some additional illustrations to complement the architecture shown in Figure 1. In particular, the details of the Generator and Discriminator and the physical meaning of the varying inputs was difficult to follow without directly referencing other papers. I also suspect that Figure 1 has some typos, such as the input to the Generator.

2) Overall, the writing was good and generally easy to read; however, the manuscript would benefit from another revision. There are still several typos and grammatical errors that would benefit from proofreading.

3) Should Eq 16 be RMSE, not SSIM?

**Reviewer #2: ** The authors present TomoGRAF, an innovative approach leveraging ultra-sparse projections to achieve high-quality 3D CT volume reconstruction, marking a significant advancement in X-ray physics and CT imaging. This work introduces the first universal framework for image-guided radiotherapy and interventional radiology, offering substantial clinical utility. While the manuscript demonstrates considerable promise for publication in PLOS ONE, the authors should address the following points prior to final acceptance.

1. Most NeRF methods rely on 2D supervision, while this paper mentions using 3D CT as the supervision signal, but does not explicitly explain how to ensure the effectiveness of 3D supervision during training. How can 3D supervision improve fidelity? It is suggested to add relevant ablation experiments or analysis to demonstrate the specific impact of 3D supervision on the final reconstruction quality.

2. The paper claims to model X-ray attenuation, but it is unclear whether scattering, beam hardening, or noise were considered. These factors are crucial for real CT simulations.

3. The paper only mentions two data augmentation techniques - random flipping and rotation. Were other data augmentation methods employed, such as noise injection or varying SID/SAD, to improve generalization capability?

4. For the loss functions in Equation (11) and Equation (12), how were the parameters α and β determined? Were ablation experiments conducted to ensure they are optimal values? For the loss function in Equation (13), how were the parameters γ, δ, and ε determined? And how were θ and ϕ determined in the evaluation metrics?

5. In the experiments, why was the peak signal-to-noise ratio (PSNR) = 25 set as the stopping threshold? Why not choose a higher PSNR value?

6. During the experimental process, your model's baseline performance was established using a single AP view. To determine model performance, reconstructions were additionally performed with 1, 2, 5, and 10 views. The view angles were specified as follows: for 1-view reconstruction, the AP view was used as reference; for 2-view reconstruction, AP and lateral views were used for inference; for 5-view reconstruction, starting from the AP view, every 72° rotation was applied to cover the full 360°; for 10-view reconstruction, starting from the AP view, every 36° rotation was applied to cover the full 360°. However, in the Discussion section, the statement "Meanwhile, TomoGRAF, besides 1-view referencing, can leverage additional X-ray views at arbitrary angles" appears to lack strong experimental support, since the experimental procedure in this study clearly defined view angles with uniform angular intervals, which seems inconsistent with "arbitrary angles". We recommend either: (1) adding experiments to demonstrate the value of arbitrary angles (e.g., using only the AP view and its adjacent angles) for TomoGRAF during inference, or (2) rephrasing this statement to better align with the actual experimental conditions.

7. TomoGRAF requires fine-tuning during the inference phase. For specific fine-tuning, the trained prior model is optimized under the supervision of the patient's 2D sparse view projections to adapt to new patient anatomies. Does this mean that during the inference phase, fine-tuning is required for projection images from every angle used? If so, this would lead to increased inference time, and when more views are used for inference, the time would increase accordingly. The paper mentions that single-view reconstruction takes approximately 344 seconds, while each additional view roughly doubles the reconstruction time, which appears relatively slow for practical applications. Can the network be improved to enhance its computational efficiency?

8. In the quantitative analysis, this paper employed SSIM, PSNR, and RMSE metrics, but it did not thoroughly discuss in the qualitative analysis whether these metrics can reflect clinical diagnostic requirements. In clinical applications, the accuracy of CT images is crucial. Have you considered inviting clinicians to evaluate the generated predicted CT images and assess their feasibility?

9. The pseudocode of the Siddon algorithm (Appendix 1) serves as an important methodological supplement. Please further clarify its specific implementation details in TomoGRAF, such as how it integrates with the fully connected network - does the network directly learn attenuation coefficients, or is this achieved through post-processing?

6. PLOS authors have the option to publish the peer review history of their article (what does this mean? ). If published, this will include your full peer review and any attached files.

**Do you want your identity to be public for this peer review?** For information about this choice, including consent withdrawal, please see our Privacy Policy .

Reviewer #1: No

Reviewer #2: No

---

## [Author Response · Author response to Decision Letter 1]

3 Jul 2025

Q: Most NeRF methods rely on 2D supervision, while this paper mentions using 3D CT as the supervision signal but does not explicitly explain how to ensure the effectiveness of 3D supervision during training. How can 3D supervision improve fidelity? It is suggested to add relevant ablation experiments or analysis to demonstrate the specific impact of 3D supervision on the final reconstruction quality.

A: We thank the reviewer for highlighting the importance of analyzing the effect of 3D supervision. Compared to prior NeRF-based approaches such as MedNeRF, our TomoGRAF framework introduces two major enhancements to improve reconstruction fidelity:

a more physically accurate forward model using Siddon’s ray tracing, and

the incorporation of 3D supervision from paired CT volumes during training.

These two design choices work in tandem to enhance anatomical consistency, especially under ultra-sparse-view constraints. Since our model ultimately generates 3D volumes at inference time, incorporating 3D supervision during training is both intuitive and effective—it aligns the learning objective with the test-time task and enables the model to learn more robust and structured anatomical priors.

We agree that many existing NeRF-based medical imaging frameworks rely on 2D supervision primarily due to the unavailability of paired 3D data for objects viewed under natural lights, rather than by design choice. In contrast, TomoGRAF is trained on paired 2D–3D data, giving the unique advantage of X-ray imaging in visualizing complex 3D anatomical structures.

While we agree that an ablation study isolating the impact of 3D supervision would be informative, the current manuscript, which is already lengthy, includes extensive evaluations across various sparse-view configurations. To maintain focus and clarity, we limited the scope to demonstrating the feasibility and effectiveness of the proposed framework. A deeper exploration of supervision strategies and architectural contributions should belong to a separate future study.

Q: The paper claims to model X-ray attenuation, but it is unclear whether scattering, beam hardening, or noise were considered. These factors are crucial for real CT simulations.

A: We thank the reviewer for raising this important point. Our current implementation of the forward model uses Siddon’s ray tracing algorithm, which simulates X-ray attenuation as line integrals through a voxelized volume. This approach assumes a monochromatic source and neglects scattering, beam hardening, and noise. While this simplified model is commonly used in simulation studies due to its computational efficiency and clarity, we agree that these effects are relevant to actual image quality. Future work will incorporate more realistic forward models that simulate polychromatic spectra, scatter, and noise characteristics in sparse reconstruction using actual projection X-ray images. That said, the focus of this study is to evaluate theoretical potential of TomoGRAF under ideal conditions.

Q: The paper only mentions two data augmentation techniques - random flipping and rotation. Were other data augmentation methods employed, such as noise injection or varying SID/SAD, to improve generalization capability?

A: We thank the reviewer for this thoughtful question. In this work, we employed random flipping and rotation as basic data augmentation strategies. We did not apply noise injection or vary geometric parameters such as SID/SAD during training, for the following reasons:

SID/SAD Parameters: These values were extracted directly from each patient’s DICOM metadata and therefore naturally vary across the cohort, reflecting realistic acquisition conditions without the need for synthetic augmentation.

Noise Injection: Our training cohort includes over 1,000 patients, each contributing multiple view angles and patches for supervision, offering substantial variability in anatomy, pose, and appearance. Since our method is designed to learn a universal prior during training—rather than rely on patient-specific fine-tuning or enhance the robustness of a CNN inference model—we determined that additional noise injection was unnecessary.

Purpose of the Prior: Since the prior is trained to capture general anatomical structure across patients and is later fine-tuned to sparse-view test data, our focus was on extracting shared structural features rather than simulating acquisition-specific degradations (which are better handled during test-time optimization).

That said, we agree that investigating the effect of more aggressive augmentations (including noise) could be a direction for further robustifying the prior, especially when adapting TomoGRAF to real clinic which requires institution-specific fine tuning on the prior. The additional augmentation is unnecessary for the current study which is based on relatively large and diverse datasets.

Q: For the loss functions in Equation (11) and Equation (12), how were the parameters α and β determined? Were ablation experiments conducted to ensure they are optimal values? For the loss function in Equation (13), how were the parameters γ, δ, and ε determined? And how were θ and φ determined in the evaluation metrics?

A: The parameters α, ϕ, γ, δ, and ξ were not used in Equations (11-12). The parameters ϕ, α and ξ were mentioned and defined in Page 12 as “Overall, the generator G takes x-ray source setup matrix K, view direction (pose) ξ=(θ,ϕ) , 2D sampling pattern v, shape code z_sh∈R^(M_s ) and appearance code z_a∈R^(M_a ) as input” as highlighted in green in the manuscript, where ϕ is the elevation angel, and θ is the azimuthal angel of view position pose ξ, and appearance code z_a is a latent vector that encodes view-dependent features and is optimized by the model during training. The parameter δ were mentioned in Page 12 as “The output is the material density δ in the corresponding x, where ϑ represents the network parameters and z_sh~p_sh and z_a~p_a with p_sh and p_a drawn from standard Gaussian distribution.” (highlighted in green in the manuscript), where δ represents the material density within the 3D CT volume which needs to be learnt by the trained model. The parameter γ was defined in page 14 as “Where L_x and L_ξ represent the latent codes of x and ξ, M_sh and M_a define the shape and appearance codes with z_sh∈R^(M_sh ) and z_a∈R^(M_a ), and γ(∙) represents positional encoding.” (highlighted in green in the manuscript), where γ(∙) represents positional encoding of the model. We did not use β in Equation (11-13) and throughout the manuscript.

Additionally, in Equations (11-13), hyperparameters λ_1, λ_2 〖,λ〗_3,〖 λ〗_4 and λ_5 are used to control the relative contributions of different loss terms (Equations 11–13) and metric components. These values were empirically chosen based on standard practices in the literature and preliminary experiments to ensure training stability and reasonable convergence.

Q: In the experiments, why was the peak signal-to-noise ratio (PSNR) = 25 set as the stopping threshold? Why not choose a higher PSNR value?

A: We appreciate the reviewer’s question. The PSNR (a hyperparameter) = 25 stopping threshold was selected based on empirical observations during test-time optimization. In our ultra-sparse view setup (e.g., 1–5 projection views), we found that PSNR values above 25 already corresponded to visually and structurally meaningful reconstructions. Setting a higher threshold (e.g., PSNR ≥ 30) offered marginal improvements while significantly increasing the computational burden and risk of overfitting to noise or limited view information. Moreover, our goal was not to achieve the maximum PSNR of the model inference to the referenced sparse views, but rather to perform efficient fine-tuning sufficient for realistic anatomical rendering. We found that PSNR ≈ 25 served as a practical and consistent early stopping criterion across different test cases. Nonetheless, we agree that adaptive or dynamic stopping strategies based on perceptual metrics can be explored in future work for deploying TomoGRAF into clinics.

Q: During the experimental process, your model's baseline performance was established using a single AP view. To determine model performance, reconstructions were additionally performed with 1, 2, 5, and 10 views. The view angles were specified as follows: for 1-view reconstruction, the AP view was used as reference; for 2-view reconstruction, AP and lateral views were used for inference; for 5-view reconstruction, starting from the AP view, every 72° rotation was applied to cover the full 360°; for 10-view reconstruction, starting from the AP view, every 36° rotation was applied to cover the full 360°. However, in the Discussion section, the statement "Meanwhile, TomoGRAF, besides 1-view referencing, can leverage additional X-ray views at arbitrary angles" appears to lack strong experimental support, since the experimental procedure in this study clearly defined view angles with uniform angular intervals, which seems inconsistent with "arbitrary angles". We recommend either: (1) adding experiments to demonstrate the value of arbitrary angles (e.g., using only the AP view and its adjacent angles) for TomoGRAF during inference, or (2) rephrasing this statement to better align with the actual experimental conditions.

A: We thank reviewer for pointing this out. We have revised our statement in Discussion to “In addition to single-view referencing, TomoGRAF is capable of incorporating multiple X-ray views from diverse directions, as demonstrated in this study using uniformly distributed acquisition angles.”

Q: TomoGRAF requires fine-tuning during the inference phase. For specific fine-tuning, the trained prior model is optimized under the supervision of the patient's 2D sparse view projections to adapt to new patient anatomies. Does this mean that during the inference phase, fine-tuning is required for projection images from every angle used? If so, this would lead to increased inference time, and when more views are used for inference, the time would increase accordingly. The paper mentions that single-view reconstruction takes approximately 344 seconds, while each additional view roughly doubles the reconstruction time, which appears relatively slow for practical applications. Can the network be improved to enhance its computational efficiency?

A: We appreciate the reviewer’s concern regarding inference efficiency. To clarify, fine-tuning is not performed independently for each projection view. Instead, during test-time optimization, random image patches from all available views are sampled and fed into the model in a unified optimization loop. As the number of input views increases, the model benefits from more diverse supervision, which may slightly increase the number of iterations needed to converge, but this increase is sublinear (as demonstrated in Figure 1 in this response letter) rather than linear. In our experiments (Table 2), we observed that while reconstruction time increases with additional views, the marginal cost per view decreases, given improved convergence behavior.

We agree that inference speed is a crucial aspect for practical deployment. Therefore, strategies to improve computational efficiency have been thoroughly discussed in the Limitations section of the original manuscript as “First, TomoGRAF requires further fine-tuning at the inference stage, which increases the reconstruction time (1-view at 344.25±10.32 s and 2-view at 719.46±26.78 s). The time further increases with inference using more views. Significant acceleration is desired for online procedures such as motion adaptive radiotherapy (59). Model compression techniques such as network pruning (60) and quantization (61) can decrease computational complexity while maintaining accuracy. Additionally, hardware acceleration via TensorRT (62) optimization or specialized processors (e.g., FPGAs (63), TPUs (64)) could also potentially improve the inference speed. Architecturally, incorporating efficient neural representations (e.g., lightweight MLPs (65) or hash-based encoding (66)) and adaptive sampling (67) methods could reduce computational overhead by prioritizing critical regions. Future work will explore these optimizations to improve TomoGRAF’s feasibility for real-time clinical applications, which would be essential for interventional procedures.” and the content has been highlighted in green in the manuscript.

Figure 1: Relationship between TomoGRAF inference time and number of referenced sparse views.

Q: In the quantitative analysis, this paper employed SSIM, PSNR, and RMSE metrics, but it did not thoroughly discuss in the qualitative analysis whether these metrics can reflect clinical diagnostic requirements. In clinical applications, the accuracy of CT images is crucial. Have you considered inviting clinicians to evaluate the generated predicted CT images and assess their feasibility?

A: We thank the reviewer for highlighting this important consideration. We agree that conventional quantitative metrics such as SSIM, PSNR, and RMSE, while commonly used in the literature, do not fully capture the diagnostic relevance of reconstructed CT images. In this study, our focus was on establishing a technical proof-of-concept for the TomoGRAF framework, and as such, we did not incorporate clinical reader evaluations. That said, we acknowledge the value of involving clinical experts in future evaluations, especially as we move toward applying this framework to real patient data. We added a note in the Discussion section to reflect this important point and outline plans for future clinical validation as “Lastly, while the current study primarily focuses on demonstrating the technical feasibility of ultra-sparse view reconstruction of the proposed TomoGRAF framework, we recognize that conventional quantitative image quality metrics may not fully capture the clinical utility of reconstructed images. As a future direction, incorporating clinical evaluation, such as qualitative scoring by radiologists or task-based diagnostic assessment, will be informative to assess the real-world applicability and reliability of TomoGRAF in clinical practice.”

Q: The pseudocode of the Siddon algorithm (Appendix 1) serves as an important methodological supplement. Please further clarify its specific implementation details in TomoGRAF, such as how it integrates with the fully connected network - does the network directly learn attenuation coefficients, or is this achieved through post-processing?

A: We thank the reviewer for this insightful question. Siddon’s ray tracing algorithm serves as the forward projection operator within the TomoGRAF framework, replacing the original ray tracing in NeRF method for natural lights to compute line integrals through the reconstructed volume.

Specifically, the fully connected network directly outputs voxel-wise attenuation coefficients that represent the 3D volume. Siddon’s algorithm then integrates these coefficients along rays corresponding to the given projection views to produce synthetic 2D projections. This forward projection step is fully differentiable and embedded within the network training loop, allowing end-to-end optimization. There is no separate post-processing step for attenuation coefficients; the network learns to represent the volume implicitly, and Siddon’s algorithm models the physics of X-ray projection during

---

## [Decision Letter · Decision Letter 1]

1 Aug 2025

TomoGRAF: An X-Ray Physics-Driven Generative Radiance Field Framework for Extremely Sparse View CT Reconstruction

PONE-D-25-15335R1

Dear Dr. Sheng,

We’re pleased to inform you that your manuscript has been judged scientifically suitable for publication and will be formally accepted for publication once it meets all outstanding technical requirements.

Kind regards,

Zhentian Wang, Ph.D.

Academic Editor

PLOS ONE

Additional Editor Comments (optional):

Both reviewers have confirmed that their comments have been addressed in the revision.

Reviewers' comments:

Reviewer's Responses to Questions

**Comments to the Author**

1. If the authors have adequately addressed your comments raised in a previous round of review and you feel that this manuscript is now acceptable for publication, you may indicate that here to bypass the “Comments to the Author” section, enter your conflict of interest statement in the “Confidential to Editor” section, and submit your "Accept" recommendation.

Reviewer #2: All comments have been addressed

2. Is the manuscript technically sound, and do the data support the conclusions?

Reviewer #2: Yes

3. Has the statistical analysis been performed appropriately and rigorously? 

Reviewer #2: N/A

4. Have the authors made all data underlying the findings in their manuscript fully available?

Reviewer #2: Yes

5. Is the manuscript presented in an intelligible fashion and written in standard English?

Reviewer #2: Yes

6. Review Comments to the Author

Reviewer #2: (No Response)

7. PLOS authors have the option to publish the peer review history of their article (what does this mean? ). If published, this will include your full peer review and any attached files.

**Do you want your identity to be public for this peer review?** For information about this choice, including consent withdrawal, please see our Privacy Policy .

Reviewer #2: No

---

## [Editor Report · Acceptance letter]

PONE-D-25-15335R1

PLOS ONE

Dear Dr. Sheng,

I'm pleased to inform you that your manuscript has been deemed suitable for publication in PLOS ONE. Congratulations! Your manuscript is now being handed over to our production team.

Kind regards,

on behalf of

Prof. Zhentian Wang

Academic Editor

PLOS ONE